# Exploring hydrophilic 2,2-di(indol-3-yl)ethanamine derivatives against *Leishmania infantum*

Alessia Centanni[1][☯], Aurora Diotallevi[2][☯], Gloria Buffi[2], Diego Olivieri[2], Nuno Santarém[3,4], Antti Lehtinen[1], Jari Yli-Kauhaluoma[1], Anabela Cordeiro-da-Silva[3,4], Paula Kiuru[1]*, Simone Lucarini[2]*, Luca Galluzzi[2]

**1** Drug Research Program, Division of Pharmaceutical Chemistry and Technology, Faculty of Pharmacy, University of Helsinki, Helsinki, Finland, **2** Department of Biomolecular Sciences, University of Urbino Carlo Bo, Urbino (PU), Italy, **3** Instituto de Investigação e Inovação em Saúde da Universidade do Porto, R. Alfredo Allen, Porto, Portugal, **4** Departamento de Ciências Biológicas, Faculdade de Farmácia da Universidade do Porto (FFUP), Porto, Portugal

☯ These authors contributed equally to this work.
* paula.kiuru@helsinki.fi (PK); simone.lucarini@uniurb.it (SL)

**Data Availability Statement:** All relevant data are within the manuscript and its Supporting Information files.

## Abstract

Herein we report the design and the synthesis of a library of new and more hydrophilic bisindole analogues based on our previously identified antileishmanial compound URB1483 that failed the preliminary *in vivo* test. The novel bisindoles were phenotypically screened for efficacy against *Leishmania infantum* promastigotes and simultaneously for toxicity on human macrophage-like THP-1 cells. Among the less toxic compounds, eight bisindoles showed IC$_{50}$ below 10 µM. The most selective compound **1h** (selectivity index = 10.1, comparable to miltefosine) and the most potent compound **2c** (IC$_{50}$ = 2.7 µM) were tested for their efficacy on *L. infantum* intracellular amastigotes. The compounds also demonstrated their efficacy in the *in vitro* infection model, showing IC$_{50}$ of 11.1 and 6.8 µM for **1h** and **2c**, respectively. Moreover, **1h** showed a better toxicity profile than the commercial drug miltefosine. For all these reasons, **1h** could be a possible new starting point for hydrophilic antileishmanial agents with low cytotoxicity on human macrophage-like cells.

## Introduction

Leishmaniasis is a neglected tropical disease that affects up to 1 million people every year. There are three main forms of the disease: cutaneous (CL, the most common), mucocutaneous (MCL), and visceral (VL), also known as kala-azar [1, 2]. VL is the most serious form and is fatal if untreated, causing up to 40000 deaths worldwide each year [3]. Leishmaniasis is caused by more than 20 different species of protozoan parasites, which are transmitted by the bite of infected female sandflies. Approximately 70 mammalian species, including humans, are natural reservoir hosts of *Leishmania* parasites [1]. Leishmaniasis threatens 350 million people in more than 90 countries across the globe, mainly in Africa, Asia, and Latin America [2]. In recent years, also due to climate change, the endemic regions for leishmaniasis have increased

**Funding:** This study was supported by a University of Urbino grant "DISB_GALLUZZI_PROG_SIC_ALIMENTARE_2021", PNRR MUR (Italian Ministry of University and Research) project ECS_00000041-VITALITY - CUP J13C22000430001, and EU COST (European Cooperation in Science and Technology) Action 21111 OneHealthDrugs. NS was funded by the Portuguese Foundation for Science and Technology through the Scientific Employment Stimulus programme, with individual grant 2021.04285.CEECIND.

**Competing interests:** The authors have declared that no competing interests exist.

to now include nontropical areas, such as Mediterranean Europe, where the most common forms of leishmaniasis are zoonotic VL and CL, both caused by *L. infantum* [4].

Currently, no valid vaccine against the disease is on the market [5]. Moreover, all the commercially available drugs are inadequate [1]. First-line treatments, such as meglumine antimoniate, are often ineffective due to the presence of resistant parasite strains. Amphotericin B (AmB) is effective against antimonial-resistant strains; however, it causes profound nephrotoxicity. AmBisome® (AmB liposomal formulation) shows a better toxicity profile, but treatments are expensive, and it requires parenteral administration [6]. Pentamidine is also efficacious against pentavalent antimony-resistant protozoan parasites; however, its use has declined due to its several side effects (nephrotoxicity, myocarditis, diabetes mellitus, hypoglycemia, hypotension, and fever), route of administration (injection) and rapidly emerging resistance. Paromomycin, an old aminoglycoside antibiotic, is now used in the treatment of leishmaniasis (repurposing strategy). Although it is the cheapest drug on the market, unfortunately it requires parental administration and prolonged treatment. Moreover, its use is limited due to drug resistance development, region-dependent efficacy, as well as common side effects such as ototoxicity and liver dysfunction. Miltefosine is a potent oral drug, but it is teratogenic, expensive, requires a prolonged treatment regimen, and has high resistance potential [1].

Since the currently available therapeutic options show serious drawbacks, interesting drug candidates are emerging, characterized by shorter treatment, low toxicity, region-independent effectiveness, and low costs. Several new anti-leishmanial lead compounds have been discovered [1, 7–11]. Among them, very few bisindoles were reported [12–17], including URB1483 (Fig 1), a pyrrole-bisindole derivative that emerged from a phenotype screening of an azole-bisindole chemical library in terms of efficacy on *L. infantum* promastigotes and amastigotes

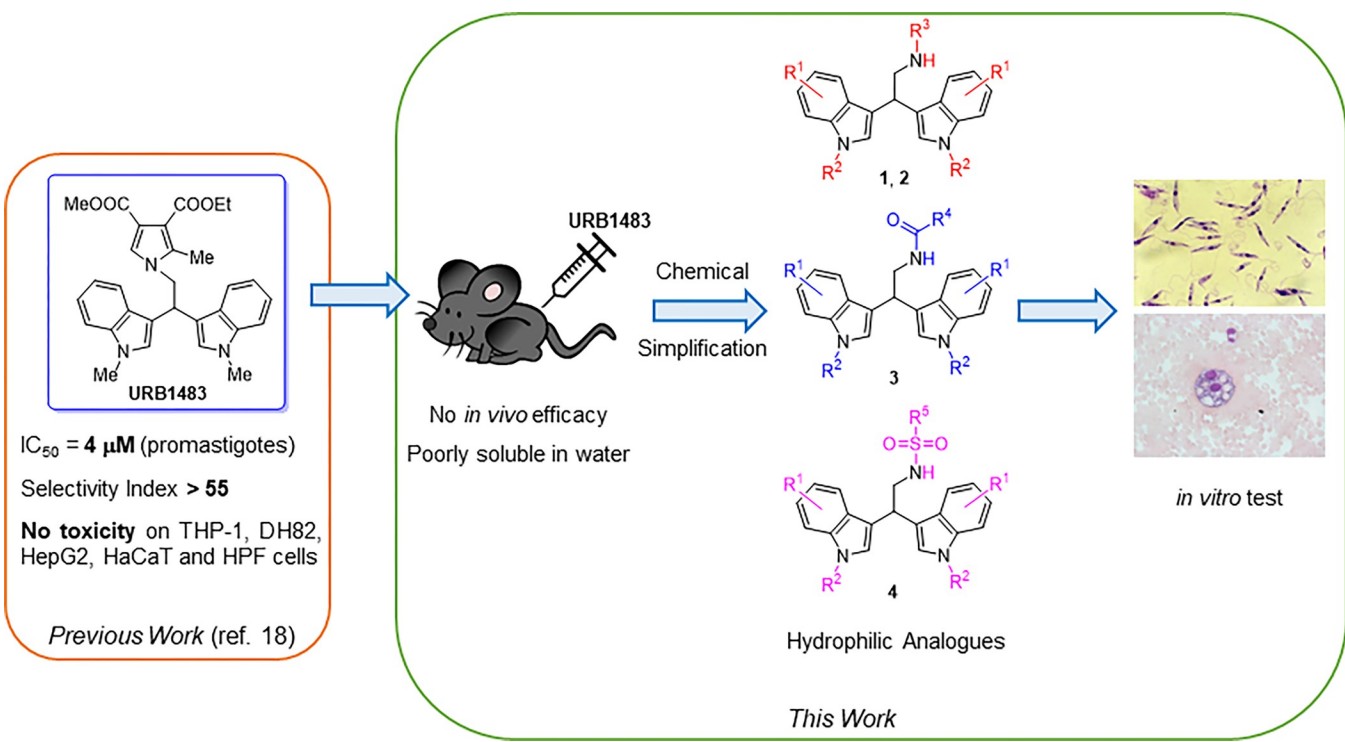

**Fig 1. Objectives of this work and chemical simplification of the lead compound URB1483 for the design of the three classes of 2,2-di(indol-3-yl) ethanamine derivatives against *L. infantum*.**

[18]. URB1483 showed good activity against the parasite ($IC_{50}$ = 3.7 ± 0.5 μM against *L. infantum* MHOM/TN/80/IPT1), and it did not affect the viability of several human cell lines, including THP-1 ($CC_{50}$ > 200 μM), with a selectivity index (SI) greater than 55. Moreover, the efficacy of URB1483 on an *in vitro* infection model was comparable to that of the commercial drug pentamidine [18].

In the present study, encouraged by the efficacy of URB1483 against *L. infantum* promastigotes and intracellular amastigotes along with its non-toxicity in mammalian cells [18], a preliminary *in vivo* experiment was performed in a mouse infection model. However, URB1483 was ineffective compared to miltefosine, probably due to poor water solubility. Therefore, four classes (**1**–**4**) of more hydrophilic analogues of URB1483 with simplified central pyrrole moieties were designed and synthesized (Fig 1). The resulting 26 compounds were screened for efficacy against *L. infantum* promastigotes and for toxicity on human macrophage-like cells. The two most effective compounds were tested for their efficacy on *in vitro L. infantum* infection model.

## Materials and methods

### *In vivo* studies

**Ethical statement.**    All experiments were approved by the I3S Animal Welfare and Ethics Review Body and are in accordance with the Portuguese National Authority for Animal Health (DGAV) guidelines, according to the statements on the directive 2010/63/EU of the European Parliament and of the Council. DGAV approved the animal experimentation presented in this manuscript under the license DGAV number 25268/2013-10-02. All experiments were carried out at the I3S Animal Facility by trained and officially qualified personnel.

**BALB/c mice infection and treatment.**    The compound URB1483 was selected for the *in vivo* evaluation on a BALB/c mice infection model. Six- to seven-week-old BALB/c mice were purchased from Charles River Laboratories (Barcelona, Spain) or the I3S (Porto, Portugal) animal facility. Luciferase-expressing *L. infantum* (MHOM/MA/67/ITMAP-263) axenic amastigotes [19] were cultured in complete cell-free medium MAA/20 [20] at 37˚C with 5% $CO_2$. Five-day-old cultures of amastigotes were washed twice with PBS and resuspended at a concentration of $1 \times 10^9$ parasites/Ml [21]. A total of 17 mice were infected with 100 μL of parasite suspension containing $1 \times 10^8$ parasites. The injection was performed intra-venously (IV) in the tails of mice, and treatments started 21 days post-infection for 10 consecutive days. Formulation studies are reported in the Supporting Information (S1 Table in S1 File). From these investigations emerged that the maximum injectable concentration of URB1483 is 5 mg/kg (1.25 mg/mL; mouse weight = 25 g; injection volume = 100 μL), using a 60% solution of PEG 400 in water. In detail, the injectable solution was prepared by dissolving 1.00 mg of URB1483 in 40 μL of DMSO, 480 mg of PEG 400, and distilled water to reach a final volume of 800 μL [22–24]. The injection solution was freshly prepared every day. Four mice were treated with PEG 400 solution without compounds and four with miltefosine (stock solution 2 mg/mL in distilled water), as negative and positive controls, respectively. The administration took place intraperitoneally (IP) with 100 μL/mouse at a concentration of 5 mg/kg. The treatment and control groups were as follows:

- 5 uninfected mice (healthy)

- 5 infected and untreated mice

- 4 mice infected and treated with 10 mg/kg/day miltefosine oral administration (per os) (positive control)

- 4 mice infected and treated with 5 mg/kg/day URB1483 IP

- 4 mice infected and treated with vehicle only (3 g/kg/day of PEG 400) IP (negative control)

***In vivo* evaluation of the anti-amastigote activity of URB1483.** Mice well-being-assessment was checked every two days, in terms of weight control and the presence of stress signs. After ten days of treatment, mice were anesthetized with 2.5% isoflurane and euthanized with cervical dislocation. Spleen, liver, and femur bone marrow were removed for parasite load calculation. Spleen and liver were weighted, and bone marrow cells were counted with automatic cell counter EVE™ (NanoEnTek, Seul, Korea). Organ homogenates (1 mg of spleen and 10 mg of liver) and $10^5$ bone marrow cells were subjected to serial dilutions in triplicate in Schneider's Insect medium (supplemented with 10% heat-inactivated fetal bovine serum (FBS), 200 U/mL penicillin/streptomycin, 6 μg/mL phenol red and 5 mM 4-(2-hydroxyethyl)-1-piperazineethanesulfonic acid (HEPES) in 96-well microtitration plates. The plates were incubated at 27°C for 15 days, and each well was inspected for the presence or absence of promastigotes and checked to find the final dilution for which the well contained at least one parasite [25]. The parasite burden (number of parasites/g of organ) was calculated as follows: [(geometric mean of reciprocal titer from each triplicate/mass of homogenized organ) × reciprocal fraction of the homogenized organ inoculated into the first well]. For bone marrow, the number of parasites was calculated per $10^5$ cells using the formula: [(geometric mean of reciprocal titer from each triplicate/number of cells) × reciprocal fraction of the cells inoculated into the first well]. For graphical representation, log transformation was applied to the parasite load to reduce right skewness.

## Chemistry

All reactions were carried out using commercially available starting materials and used without further purification unless otherwise stated. All reagents, solvents, and dry solvents (acetonitrile, dichloromethane, and toluene) were purchased from TCI (Zwijndrecht, Belgium) or Fluorochem (Hadfield, UK) or Alpha Aesar (Ward Hill, Massachusetts, USA) or Sigma-Aldrich/Merck Life Science (Europe). $^1$H and $^{13}$C NMR spectra were acquired on a Bruker Ascend Avance III HD 400 and Bruker Avance 400 NMR spectrometers ($^1$H: 400MHz, $^{13}$C: 101 MHz), and analyzed using the TopSpin 1.3 (2013) or MestReNova x64 software packages. Chemical shifts ($\delta$) are given in parts per million (ppm) relative to the NMR reference solvent signals (CDCl$_3$: 7.26 ppm and 77.16 ppm; CD$_3$OD: 3.31 ppm and 49.00 ppm, (CD$_3$)$_2$CO: 2.05 ppm and 29.84 ppm; DMSO-$d_6$: 2.50 ppm and 39.52 ppm). Multiplicities are indicated by s (singlet), d (doublet), dd (doublet of doublets), ddd (doublet of doublets of doublets), t (triplet), q (quartet), and m (multiplet). Visual features of peaks including broad (br) or apparent (app) are also indicated. The coupling constants *J* are quoted in hertz (Hz). Flash SiO$_2$ column chromatography was performed with automated high performance flash chromatography, Biotage Isolera™Spektra Systems with ACI™ and Assist (ISO-1SW Isolera One) equipped with a variable UV-VIS (200–800 nm) photodiode array using SNAP KP-SIL, Ultra/NH or Sfär Duo/C18 5g, 10 g or 25 g cartridges, and the indicated mobile phase gradient. Classical column chromatography purifications were performed under "flash" conditions using Merck 230–400 mesh silica gel. TLC analyses were performed using Merck (Silica gel 60-F254) TLC plates and visualized by exposure to ultraviolet light and/or by exposure to an aqueous solution of ceric ammonium molybdate, potassium permanganate or ninhydrin and plate heating. ESI-MS spectra were recorded with a Waters Micromass ZQ spectrometer in a negative or positive mode using a nebulizing nitrogen gas at 400 L/min and a temperature of 250°C, cone flow 40

mL/min, capillary 3.5 kV and cone voltage 60 V. LC-MS and HRMS-spectra were recorded using a Waters Acquity UPLC®-system (with Acquity UPLC® BEH C18 column, 1.7 μm, 50 mm × 2.1 mm, Waters) with Waters Synapt G2 HDMS with the ESI (+), high resolution mode. The mobile phase consisted of $H_2O$ (A) and acetonitrile (B) both containing 0.1% HCOOH. Microwave syntheses were performed in sealed tubes using Biotage Initiator+ instrument equipped with an external IR sensor and the set temperature mode was used. The purity of compounds URB1483, **1a**, **1b**, **1h**, **1i** and **4b** were analyzed on a ThermoQuest (Rodano, Italy) FlashEA 1112 elemental analyzer for C, H, and N. The percentages found were within ±0.5% of the theoretical values. Otherwise, LC-MS analysis was utilized to assess the purity. All the tested compounds were >95% pure (see the Supporting Information).

## Synthesis of ethyl 1-[2,2-bis(1-methyl-1*H*-indol-3-yl)ethyl]-2-methyl-4-(propionyloxy)-1*H*-pyrrole-3-carboxylate (URB1483)

Compound URB1483 has been synthesized as described in the literature. [18, 26] Characterization data can be found in the Supporting Information.

**General procedure A for the synthesis of bisindole 1.** The solution of *N*-(2,2-dimethoxyethyl)-2,2,2-trifluoroacetamide (0.4–13 mmol, 1 equiv) was dissolved in dry acetonitrile (1 mL/mmol) in a microwave (MW) tube under an argon atmosphere. The appropriate indole (2 equiv) and diphenyl phosphate (DPP, 0.04–1.3 mmol, 0.1 equiv) were added to the solution. The mixture was heated at 120˚C under MW irradiation for 4–8 h and allowed to cool to room temperature. The mixture was extracted with dichloromethane (DCM, 30 mL) and washed with a saturated solution of $NaHCO_3$ in water (3 × 20 mL). The organic layer was washed with brine (20 mL), dried over anhydrous $Na_2SO_4$, filtered and the solvent was removed *in vacuo*. The product was purified by silica gel column chromatography using either *n*-heptane/EtOAc or DCM/MeOH gradient.

Then, the appropriate trifluoroacetamide isolated derivatives (1.0–4.5 mmol) and potassium carbonate (2.5 mmol) in MeOH:$H_2O$ (20:1) was refluxed for 1–3 h. The solvent was removed under reduced pressure. Water (30 mL) was added, and the resulting aqueous solution was extracted with DCM (3 × 30 mL). The organic layer was dried with anhydrous $Na_2SO_4$, filtered and concentrated *in vacuo*. The product was purified by silica gel column chromatography. Characterization data can be found in the Supporting Information.

**General procedure B for the synthesis of bisindole bearing secondary amines 2.** To a stirred solution of bisindole **1d** (0.3 mmol, 1 equiv) in methanol (6 mL) was added glacial acetic acid (1.38 mmol) followed by sodium cyanoborohydride (0.78 mmol) under argon atmosphere at 0˚C. The appropriate aldehyde (0.33 mmol, 1.1 equiv) in methanol (4 mL) was then added. The resulting mixture was stirred at room temperature for 2–4 h and the reaction was monitored by TLC. A 2 M solution of $Na_2CO_3$ in water (15 mL) was added to adjust the pH to 8–9, and the solvent was removed *in vacuo*. The residue was partitioned between DCM and water and the organic layer was washed with water (15 mL) and brine (15 mL), dried with anhydrous $Na_2SO_4$, filtered, and concentrated *in vacuo*. The products were purified with automated flash chromatography. Characterization data can be found in the Supporting Information.

**General procedure C for the synthesis of bisindole bearing secondary amides 3.** The selected carboxylic acid (0.2 mmol) was dissolved in anhydrous tetrahydrofuran (THF, 10 mL), followed by addition of *N*,*N*-diisopropylethylamine (DIPEA, 0.6 mmol, 3 equiv) under an argon atmosphere. 1-Ethyl-3-(3-dimethylaminopropyl)carbodiimide (EDC) hydrochloride (0.3 mmol, 2 equiv) was added at 0˚C, and the mixture was stirred at room temperature for 30 min, after which bisindole **1d** (0.2 mmol) was added. The reaction mixture was stirred for 20 h

at room temperature under argon atmosphere and was monitored by TLC. The residue was partitioned between EtOAc (20 mL) and a saturated solution of NH$_4$Cl in water (20mL) and the organic layer was extracted with EtOAc (2 × 20 mL). The combined organic layers were washed with brine (15 mL), dried with anhydrous Na$_2$SO$_4$, filtered, and concentrated *in vacuo*. The product was purified with automated flash chromatography. Characterization data can be found in the Supporting Information.

**General procedure D for the synthesis of bisindole bearing sulphonamides 4.** To a stirred solution of bisindole **1** (0.15 mmol, 1 equiv) in THF (6 mL), triethylamine (TEA) was added (2 equiv). The mixture was allowed to cool to 0˚C and the appropriate sulfonyl chloride (0.18 mmol, 1.2 equiv) was added. The mixture was stirred at room temperature for 1 h. The reaction mixture was concentrated *in vacuo*, and the crude product was purified using the automated flash chromatography. Characterization data can be found in the Supporting Information.

## *In vitro* studies

**Parasite and cell cultures.** The reference strain *L. infantum* MHOM/TN/80/IPT1 (WHO international reference strain) was purchased from ATCC (ATCC® 50134™) and cultured in Evans's Modified Tobie Medium (EMTM) at 26–28˚C. During the treatment with bisindole derivative compounds, the parasites were maintained in RPMI-PY medium as described previously [18, 27]

The human monocytic cell line THP-1 (ECACC 88081201) was cultured in RPMI-1640 medium supplemented with 10% heat-inactivated fetal bovine serum (FBS), 2 mM L-glutamine, 10 g/L non-essential amino acid, 1 mM sodium pyruvate, 100 μg/mL streptomycin, 100 U/L penicillin. Cells were maintained in a humidified incubator at 37˚C and 5% CO$_2$. All cell culture reagents were purchased from Sigma-Aldrich (St. Louis, MO, USA).

**_L. infantum_ promastigotes viability assay.** The activity of bisindole derivatives was first evaluated on late log/stationary phase *L. infantum* promastigotes resuspended in 96-well plates (100 μL/well) in complete RPMI-PY medium with a density of 2.5×10$^6$ parasites/mL. The antiparasitic activity was initially investigated with the single dose of 20 μM of each compound for 72 h at 26˚C. To analyze the dose-response, all compounds that showed an antiparasitic activity > 50% at 20 μM were further tested at scalar dilutions 1:2 or 2:3 (from 20 to 0.31 μM) on promastigotes. As positive control, the anti-leishmanial drug miltefosine (Sigma-Aldrich, Milan, Italy) was included. Each condition was carried out in duplicate. The promastigotes viability was evaluated using the CellTiter 96H aqueous non-radioactive cell proliferation assay (Promega, Madison, Wisconsin, USA), as previously reported [18]. The efficacy of selected compounds was evaluated by determining the IC$_{50}$ values using the non-linear regression curves in GraphPad Prism 8.0 (GraphPad Software, Inc., San Diego, CA, USA). The equation used for data fitting was Y = 100/(1+10^((LogIC$_{50}$-X)*HillSlope)) (hillslope not constrained), where X is equal to the log of concentration and Y is the normalized response.

**Cytotoxicity assay.** The cytotoxicity of bisindole derivative compounds was evaluated on THP-1 cells. Briefly, cells were resuspended at a density of 5×10$^6$ cells/mL, 100 μL/well were seeded in a 96-well plate and treated for 48 h with 20 ng/mL phorbol myristic acid (PMA) to induce differentiation into macrophages-like cells. After cell adhesion, selected compounds were used at 4, 20, and 100 μM, for 72 h at 37˚C. Compounds that showed cytotoxicity between 4 and 20 μM were not included in subsequent experiments. Compounds with cytotoxicity higher than 20 μM were further tested to determine CC$_{50}$ with scalar dilutions 1:2 or 2:3 (from 100 to 3.12 μM).

The negative control (not-treated cells) and the anti-leishmanial drug miltefosine were included in each experiment. Each condition was carried out in duplicate. To evaluate the

cytotoxicity of the selected compounds, the CellTiter 96H Aqueous Non-Radioactive Cell Proliferation Assay (Promega) was performed as described above. For each compound, the SI was calculated as the ratio between cytotoxicity in THP-1 and activity against *L. infantum* promastigotes.

**Anti-amastigote assay on infected cells.** The anti-amastigote activity of compounds **1h** and **2c** was evaluated in THP-1-infected cells. THP-1 cells were plated in a 96-well plate and differentiated as described above. After differentiation, cells were infected with a parasite-to-cell ratio of 10:1 for 24 h with *L. infantum* MHOM/TN/80/IPT1 promastigotes previously stained with carboxyfluorescein succinimidyl ester (CFSE) dye (Invitrogen, Waltham, Massachusetts, USA) following manufacturer's instructions. Briefly, *Leishmania* cells were washed twice with PBS to remove any trace of serum and resuspended $1 \times 10^7$/mL PBS. CFSE was then added at 5 μM final concentration, and promastigotes were incubated for 15 min at 37˚C in the dark. The labeling was stopped by adding 4 volumes of cold complete media containing 10% serum and incubating in ice for 5 min. Labeled parasites were washed three times with complete RPMI medium and used for the infection. After 24 h, the medium was replaced with fresh RPMI to remove non-internalized parasites, and cells were treated with **1h**, **2c**, or with the positive control miltefosine with scalar dilutions 1:2 and 2:3 (from 20 to 0.31 μM), for 72h. To analyze the infection, cells were washed, formaldehyde/methanol fixed, stained with Hoechst dye, and observed with a fluorescence microscope. Images were acquired (at 5× magnification) at 494 nm excitation/521 nm emission (green) and 350 nm excitation/460 emission (blue) for CFSE and Hoechst staining, respectively. Automated cell counting of single-color images was performed with ImageJ software to monitor the infection: the promastigotes count on each green image was normalized with the cell count on the corresponding blue image. The normalized value of infected non-treated cells was taken as 100%. To calculate $IC_{50}$, the percentage values of *Leishmania* cell count normalized to THP-1 cells were plotted against the $\log_{10}$ concentration (μM) of each compound.

## Statistical analysis

The statistical analysis for the *in vivo* experiments was performed by Ordinary one-way ANOVA with Dunnett's multiple comparisons test (single pooled variance). For *in vitro* studies, the evaluation of $IC_{50}$ in promastigotes and $CC_{50}$ in mammalian cells following bisindole derivatives treatment was performed by non-linear regression analysis and expressed as means and 95% confidence interval. Cytotoxicity statistical analysis on infected cells was performed using two-way ANOVA followed by Dunnett's multiple comparisons test. All statistical tests were performed using GraphPad Prism version 8 (GraphPad Software, Inc., La Jolla, CA, USA). A p-value ≤ 0.05 was considered significant.

## Results

### *In vivo* studies

Considering the results on intracellular amastigotes, the compound URB1483 was selected for *in vivo* experiments on infected BALB/c mice. Based on the literature data [22–24] a formulation study of URB1483 was performed in order to obtain an injectable system (see Supporting Information). Different biocompatible excipients, such as dimethylsulfoxide (DMSO), propylene glycol (PG), polyethylene glycol (PEG) 400, and 2-hydropropyl-β-cyclodextrin (HPBC), have been used. The best result, in terms of solubility, has been reached by employing a formulation of 60% of PEG 400 in water, which allowed an IP injectable dose of 5 mg/kg in mice (URB1483 solution 1.25 mg/mL; injection volume = 100 μL; average mouse weight = 25 g).

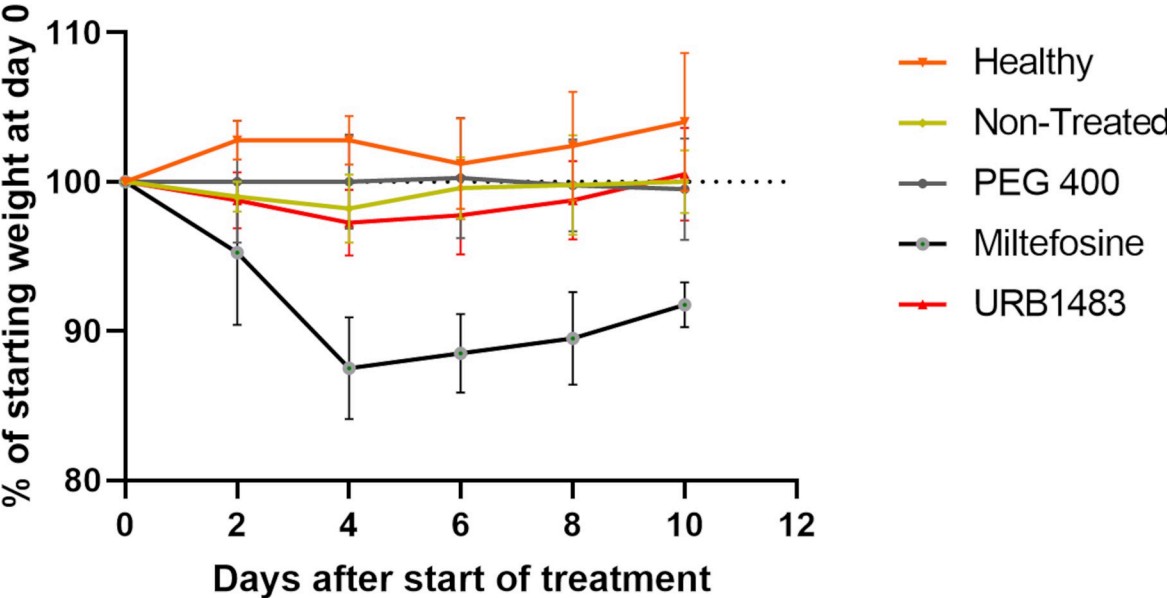

**Fig 2. BALB/c weight variation upon *Leishmania* infection and treatment.** BALB/c mice (6–8 weeks old) were infected intraperitoneally with $1\times10^8$ *L. infantum* axenic amastigotes. Three weeks after infection, the animals were treated with 5 mg/kg/day URB1483 (IP), 10 mg/kg/day miltefosine (per os), or 3 g/kg/day PEG 400 (IP) for 10 consecutive days. Every second day after the start of treatment, the weight of each mouse was determined. Each curve represents the average weight variation and standard deviation of the group at specific time points compared to the weight at the beginning of treatment.

Seventeen BALB/c mice were infected with $1\times10^8$ *L. infantum* axenic amastigotes intravenously. After 21 days post-infection, mice were treated for 10 days, as described in materials and methods. Miltefosine and PEG 400 administration were included as positive and negative controls, respectively. Every two days, mice were checked to control the weight and the presence of stress signs. The formulation administered in 60% PEG 400 was well tolerated by the mice that did not show any alteration either in physical or behavioral status. The only change was found in mice treated with miltefosine where a weight decrease of 10% occurred from the second treatment. Nevertheless, a weight loss ≤ 20% was above the defined humane endpoint (Fig 2).

After 10 days of treatment, mice were euthanized, and spleen, liver, and femur bone marrow were removed. Since *Leishmania* infection is associated with splenohepatomegaly, the weight of the organs was considered important when evaluating the effectiveness of the treatment. The infected animals presented significantly heavier spleens than the non-infected (healthy) animals. In animals treated with effective drugs, the increased spleen weight is expected to return to normal. This occurred only in the miltefosine-treated group that presented average spleen weights significantly lower to the non-treated group and comparable to the healthy group (Ordinary one-way ANOVA with Dunnett's multiple comparisons test. Single pooled variance) (Fig 3, panel A)

The data on the liver shows no differences among all the groups probably because our infection model did not originate hepatomegaly (Fig 3, panel B). Moreover, to evaluate the efficacy of the treatment to fight infection, the calculation of the parasite burden was performed using the gold standard limiting dilution assay. Organ homogenates (1 mg of spleen and 10 mg of liver) and $1\times10^5$ bone marrow cells were subjected to serial dilutions in 96-well microtitration plates and were incubated at 27°C for 15 days to find the final dilution for which the well contained at least one parasite. Results showed that only miltefosine was able to reduce the parasite burden in all the organs (more than 99.9%) (Fig 4).

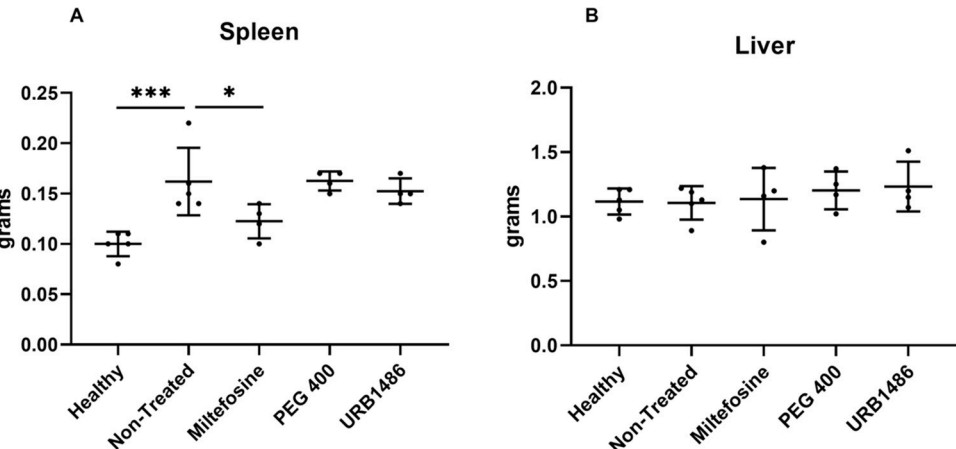

**Fig 3. Organ weight in BALB/c upon *Leishmania* infection and treatment.** BALB/c mice (6–8 weeks old) were infected intraperitoneally with $1 \times 10^8$ *L. infantum* axenic amastigotes. Three weeks after infection, the animals were treated with 5 mg/kg/day URB1483 (IP), 10 mg/kg/day miltefosine (per os), or 3 g/kg/day PEG 400 (IP) for 10 consecutive days. One day after the last treatment the mice were euthanized, and the organ weight of each individual animal was registered. Each individual dot represents the organ weight of an individual animal, the horizontal bar represents the average and associated standard deviation of the group. * p < 0.05, ***p < 0.001 (Ordinary one-way ANOVA with Dunnett's multiple comparisons test; single pooled variance).

In particular, no parasites were detected in the liver after the miltefosine treatment (considering the limit of detection of the technique) (Fig 4, panel A). Unfortunately, apart from miltefosine no other treatment was able to significantly reduce parasite burden in any organ (Ordinary one-way ANOVA with Dunnett's multiple comparisons test; single pooled variance). Thus, the treatment with URB1483 did not significantly reduce parasite burden. This is either suggestive of poor bioavailability or lack of activity *in vivo*.

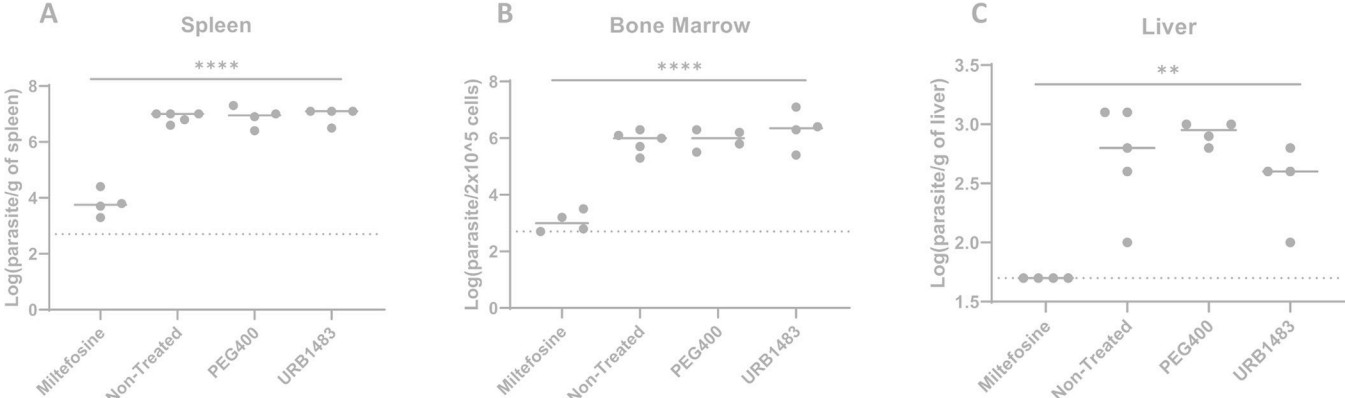

**Fig 4. *In vivo* efficacy of URB1483 in *L. infantum*-infected mice.** BALB/c mice (6–8 weeks old) were infected intraperitoneally with $1 \times 10^8$ *L. infantum* axenic amastigotes. Three weeks after infection, the animals were treated with 5 mg/kg/day URB1483 (IP), 10 mg/kg/day miltefosine (per os), or 3 g/kg/day PEG 400 (IP) for 10 consecutive days. One day after the last treatment the mice were euthanized, and parasite burden (log number of parasites/mg of organ) was evaluated by limiting dilution assay in the spleen (A), bone marrow (B), and liver (C). Each dot represents the parasite burden of an individual animal, the horizontal bar represents the median of the group. The horizontal dashed line represents the limit of detection of the technique ** p < 0.01, ****p < 0.0001 (Ordinary one-way ANOVA with Dunnett's multiple comparisons test; single pooled variance).

## Chemistry

Based on the results obtained with the *in vivo* studies on URB1483, we decided to synthesize and test more hydrophilic bisindole derivatives, which can be ideally grouped into four different series based on the substitution on the amino group (*vide infra*). In particular, the ethylamino bisindoles **1** (Scheme 2) were obtained by optimizing our earlier protocol [28], which employs heating at 80°C for 48 h (Table 1). When the reaction between 6-bromoindole **5** and *N*-(2,2-dimethoxyethyl)-2,2,2-trifluoroacetamide **6** [29] in the presence of diphenyl phosphate (DPP) was carried out under our previously optimized condition, a 0.4:1 ratio between the unreacted **5** and *N*-[2,2-bis(6-bromo-1*H*-indol-3-yl)ethyl]-2,2,2-trifluoroacetamide **7** was observed (Scheme 1 and Table 1, entry 1). When MW irradiation was employed for 4 h, an enhancement in the amount of the intermediate **7** was detected by rising the temperature from 80°C to 120°C (entries 2–4), with a 0.2:1 ratio between starting indole **5** and intermediate **7**. Finally, the selected reaction time was 4 to 8 h, since only a slight improvement was detected when the reaction was carried out for 24 h at 120°C (Table 1, entry 5). With the optimized reaction conditions in hand, the desired free amines **1a-i** were eventually obtained via MW-mediated coupling of the 5- or 6-substituted indoles and **6** (Scheme 2), followed by the basic hydrolysis of the corresponding bisindole trifluoroacetamide intermediates with potassium carbonate in good to excellent overall yields.

The other classes were synthetized from the privileged scaffold **1d**, since 2,2-bis(6-bromo-1*H*-indol-3-yl)ethanamine is a marine bisindole alkaloid (isolated from the californian tunicate *Didemnum candidum* and the new caledonian sponge *Orina* spp.) with several biological activities [30, 31]. Furthermore, 6-Br bisindole derivatives could be easily late-stage functionalized by classical cross coupling reactions (e.g., Mizoroki-Heck reaction, Stille reaction, Suzuki-Miyaura reaction), accessing to potentially new active compounds. In particular, the

**Scheme 1. Model reaction for the MW optimization of protected bisindoles 7 synthesis.**

|  | R¹ | R² |
|---|---|---|
| **1a** | H | H |
| **1b** | 6-F | H |
| **1c** | 6-Cl | H |
| **1d** | 6-Br | H |
| **1e** | 5-I | H |
| **1f** | 6-COOMe | H |
| **1g** | 5-CN | H |
| **1h** | H | Me |
| **1i** | 6-Br | Me |

**Scheme 2. Synthesis of bisindoles 1.** Regents and conditions: i) DPP, MeCN, 120°C, MW, 4–8 h; ii) K₂CO₃, MeOH, H₂O, reflux 2 h.

**Table 1. MW optimization for the 6-bromobisindole trifluoroacetamide intermediate 7 synthesis.**

| Entry | Condition | T (°C)[a] | Time (h) | Ratio[b] |
|-------|-----------|-----------|----------|----------|
| 1 | Standard heating | 80 | 48 | 0.4:1 |
| 2 | MW | 80 | 4 | 1:1 |
| 3 | MW | 100 | 4 | 0.7:1 |
| 4 | MW | 120 | 4 | 0.2:1 |
| 5 | MW | 120 | 24 | 0.1:1 |

[a] The temperatures were set constant during the MW irradiation, and the power varied between 30–75 W and the pressure was in the range 1.5–3 bar.

[b] Data is based on the ratio of the integral of **5**'s peak at 7.24 ppm compared to intermediate **7**'s peak at 7.36 ppm in CDCl$_3$. Ratios are calculated from the $^1$H-NMR spectrum of the crude mixture.

secondary amine series **2** was synthesized using reductive amination of 2,2-bis(6-bromo-1*H*-indol-3-yl)ethanamine **1d** in the presence of NaBH$_3$CN and the appropriate aldehyde (Scheme 3), bearing aryl, heteroaryl or alkyl groups (see Supporting Information). Yields from 45% to 60% were obtained for amines bearing aryl and heteroaryl groups **2b**-**g**, while the yield of iso-propylamine derivative **2a** was 83%. The amide series **3** was synthesized *via* amide coupling of the free amine **1d** with the corresponding carboxylic acids in the presence of EDC hydrochloride and DIPEA (Scheme 3). The aryl amides **3b**-**g** yields were between 40 and 53%, while a slightly better yield was achieved for the isopropylamide derivative **3a** (63% isolated yield). Additionally, three sulfonamide derivatives **4a**-**c** were synthesized in excellent yields using the

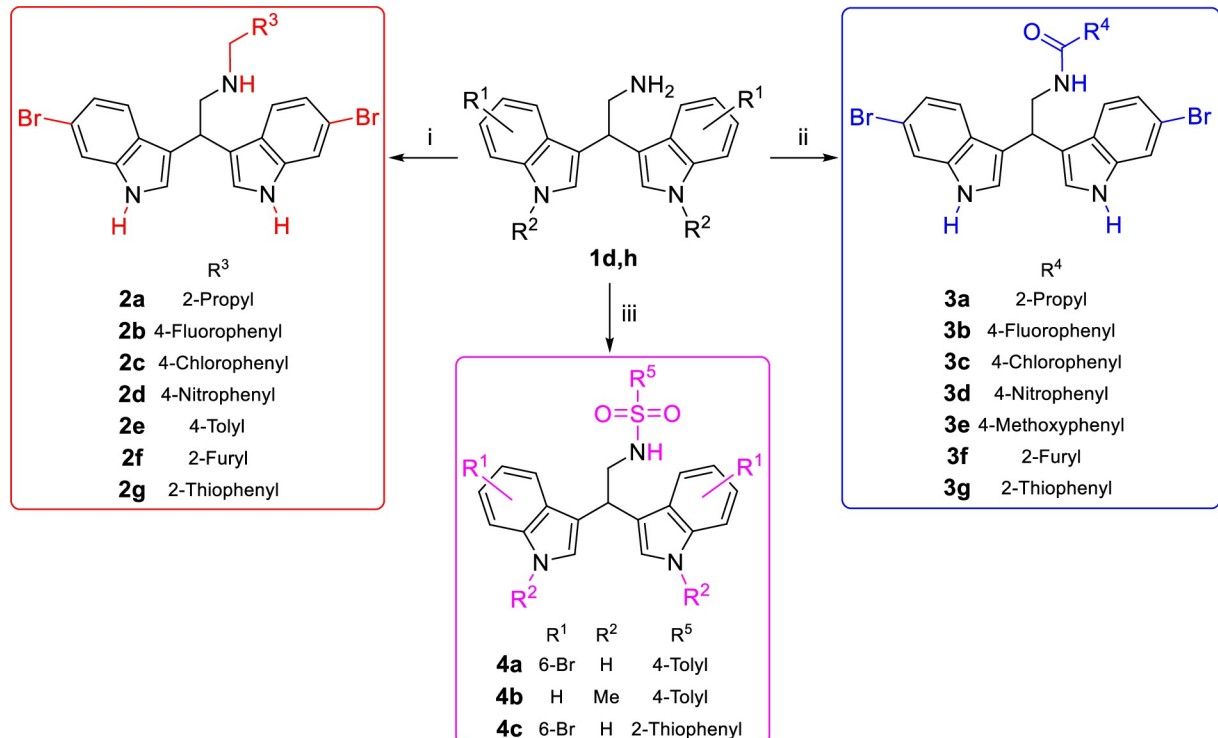

**Scheme 3. Synthesis of bisindoles 2–4** Regents and conditions: i) R$^3$-CHO, NaBH$_3$CN, AcOH, MeOH, rt, 2–4 h; ii) R$^4$-CO$_2$H, EDC·HCl, DIPEA, THF, rt, 20 h; iii) R$^5$-SO$_2$Cl, TEA, THF, rt, 2 h.

free amines **1d** or **1h** and the appropriate aryl sulfonyl chlorides in the presence of TEA as a base, as reported in Scheme 3.

### *In vitro* studies

**Test of bisindole derivatives on *L. infantum* promastigote and THP-1 cells.**   The *in vitro* anti-parasitic activity of 26 bisindole derivatives was first evaluated on *L. infantum* MHOM/ TN/80/IPT1 promastigotes at a single dose of 20 μM for 72 h. All compounds, except for **1g** and **4b**, showed an anti-leishmanial activity > 80% at 20 μM and therefore they were taken into consideration for testing on THP-1 derived-macrophages. In particular, classes **2** and **4**, showed an anti-promastigote activity >95%, with the exception of derivative **3d** (90.4%). Fourteen out of 24 compounds (i.e., **1d, 1e, 1i, 2a, 2b, 2e-g, 3a, 3c, 3d, 3f, 3g** and **4c**) showed cytotoxicity between 4 and 20 μM and therefore were not included in subsequent experiments (Table 2).

Consequently, ten active and less toxic compounds were tested on *L. infantum* MHOM/ TN/80/IPT1 promastigotes for 72 h with scalar dilution 1:2 and 2:3 (from 20 to 0.31 μM) to determine $IC_{50}$. Among them, compounds **1a** and **3e** showed $IC_{50}$ >10 μM, while all the others showed $IC_{50}$ values between 2.7 and 9.8 μM (Table 2). Therefore, the exact cytotoxicity ($CC_{50}$) of the eight most active compounds **1b, 1c, 1f, 1h, 2c, 2d, 3b,** and **4a** was evaluated on THP-1 cells as described in Methods section. The $CC_{50}$ values were between 11.7 and 41.6 μM, including miltefosine, which showed $CC_{50}$ of 36.7 μM. Notably, except for the miltefosine, the 95% confidence interval (CI) was not determined in seven out of eight compounds due to the steeper slope in dose-response curves that indicate a high potency of the compounds (S1 Fig in S1 File). Compound **1h** showed a good $IC_{50}$ against the protozoa (4.1 μM), together with the lowest toxicity on THP 1 cells (41.6 μM), therefore resulting in the highest SI (10.1), comparable to miltefosine (Table 2). On the other hand, compound **2c** is the most potent compound against *L. infantum* promastigotes ($IC_{50}$ = 2.7 μM), despite showing a relatively high toxicity on human cells. Finally, based on these results, these latter compounds **1h** and **2c** were selected for testing in intracellular amastigotes.

**Efficacy of compounds 1h and 2c on *L. infantum* intracellular amastigotes.**   Since **1h** and **2c** were found to be the most selective and effective compounds respectively against the parasites, they were selected for further experiments on intracellular amastigotes. The human monocytic THP-1 cell line infection was conducted with *L. infantum* MHOM/TN/80/IPT1 as described in Methods section. Infected macrophages were treated with compound **1h** and **2c** or miltefosine (positive control) for 72 h at scalar dilution 1:2 and 2:3 (from 20 to 0.31 μM). The analysis revealed that the infection rate was significantly reduced following the treatment with all the compounds, in a dose-dependent manner (Fig 5 and S2-S4 Figs in S1 File).

The tested compounds **1h** and **2c** showed good activity against *L. infantum* intracellular amastigotes ($IC_{50}$ = 11.1 and 6.8 μM, respectively), although miltefosine appeared to be the most effective compound ($IC_{50}$ = 1.8 μM). Bisindole **2c** confirmed to have a high cellular toxicity in the infection model (Fig 6). Nevertheless, since it showed the best activity against amastigotes, it could be a starting point to inspire a new promising class of bisindoles without bromine atoms.

## Discussion

There are numerous problems with therapeutic approaches to leishmaniasis, such as drug resistance, systemic toxicity, and high costs of production/administration, which are reflected in lower clinical success rates, suggesting an urgent need for effective new alternatives. Encouraged by excellent *in vitro* results obtained with URB1483, such as potency against *L. infantum* promastigotes and intracellular amastigotes with no quantifiable cytotoxicity in mammalian

**Table 2. Summary of bisindole derivatives activity on *L. infantum* promastigotes and THP-1 cells and corresponding SIs.** $IC_{50}$ and $CC_{50}$ values are reported as mean and 95% CI, from at least two independent experiments. Each experimental condition was conducted at least in duplicate, and miltefosine (MILT) was used a positive control.

| Cmpd | Inhibition of *L. infantum* promastigotes at 20 μM (%) | Cytotoxicity on THP-1 cells (100, 20, 4 μM) | *L. infantum* promastigotes $IC_{50}$ (μM) (95% CI) | THP-1 cells $CC_{50}$ (μM) (95% CI) | SI ($CC_{50}/IC_{50}$) |
|---|---|---|---|---|---|
| **Free amines** | | | | | |
| **1a** | 82.3 | $20 < CC_{50} < 100$ | 12.9 (11.6 to 14.6) | n.a. | |
| **1b** | 89.1 | $20 < CC_{50} < 100$ | 8.6 (7.6 to 9.7) | 40.4 | 4.7 |
| **1c** | 95.6 | $20 < CC_{50} < 100$ | 7.4 (6.8 to 8.2) | 39.4 | 5.3 |
| **1d** | 96.8 | $4 < CC_{50} < 20$ | n.a. | n.a. | |
| **1e** | 97.8 | $4 < CC_{50} < 20$ | n.a. | n.a. | |
| **1f** | 89.7 | $20 < CC_{50} < 100$ | 8.2 (7.4 to 9.2) | 40.1 | 4.9 |
| **1g** | -1.6 | n.a. | n.a. | n.a. | |
| **1h** | 98.0 | $20 < CC_{50} < 100$ | 4.1 (3.7 to 4.5) | 41.6 | 10.1 |
| **1i** | 98.0 | $4 < CC_{50} < 20$ | n.a. | n.a. | |
| **Alkyl and arylamines** | | | | | |
| **2a** | 97.8 | $4 < CC_{50} < 20$ | n.a. | n.a. | |
| **2b** | 98.2 | $4 < CC_{50} < 20$ | n.a. | n.a. | |
| **2c** | 98.1 | $20 < CC_{50} < 100$[a] | 2.7 (2.5 to 2.9) | 11.7[a] | 4.3 |
| **2d** | 95.0 | $20 < CC_{50} < 100$[a] | 3.9 (3.6 to 4.3) | 13.2 (13.1 to 13.9)[a] | 3.4 |
| **2e** | 95.8 | $4 < CC_{50} < 20$ | n.a. | n.a. | |
| **2f** | 95.2 | $4 < CC_{50} < 20$ | n.a. | n.a. | |
| **2g** | 95.5 | $4 < CC_{50} < 20$ | n.a. | n.a. | |
| **Alkyl and arylamides** | | | | | |
| **3a** | 96.7 | $4 < CC_{50} < 20$ | n.a. | n.a. | |
| **3b** | 97.7 | $20 < CC_{50} < 100$ | 9.5 (8.4 to 10.6) | 21.9 | 2.3 |
| **3c** | 97.4 | $4 < CC_{50} < 20$ | n.a. | n.a. | |
| **3d** | 90.4 | $4 < CC_{50} < 20$ | n.a. | n.a. | |
| **3e** | 95.2 | $20 < CC_{50} < 100$ | 12.2 (10.4 to 14.5) | n.a. | |
| **3f** | 97.2 | $4 < CC_{50} < 20$ | n.a. | n.a. | |
| **3g** | 96.9 | $4 < CC_{50} < 20$ | n.a. | n.a. | |
| **Aryl sulfonamides** | | | | | |
| **4a** | 96.9 | $20 < CC_{50} < 100$ | 9.8 (8.91 to 11.0) | 20.2 | 2.1 |
| **4b** | 45.7 | n.a. | n.a. | n.a. | |
| **4c** | 97.6 | $4 < CC_{50} < 20$ | n.a. | n.a. | |
| **Reference compound** | | | | | |
| **MILT** | 96.7 | $20 < CC_{50} < 100$ | 3.6 (3.3 to 4.0) | 36.7 (32.9 to 40.7) | 10.2 |

[a] Discrepancy probably due to rough determination of cytotoxicity using only three concentration points.

cells [18], a preliminary *in vivo* test on a mouse infection model was performed. In the attempt to increase the very poor water solubility of the lipophilic URB1483, a vehicle of 60% PEG 400 in water was used, this permitting a maximum dose of 5 mg/kg. Daily intraperitoneal injection of URB1483 formulation for 10 days could not reduce parasite burden in any observed organ (i.e., spleen, liver, and bone marrow), indicating therapeutic failure under these conditions. Compared with the reference drug miltefosine, used at 10 mg/kg, the absence of *in vivo* efficacy of URB1483 was probably due to its low dose resulting from its poor water solubility, and different administration route. Further studies addressing the pharmacokinetic profile of the molecule might help elucidate this outcome.

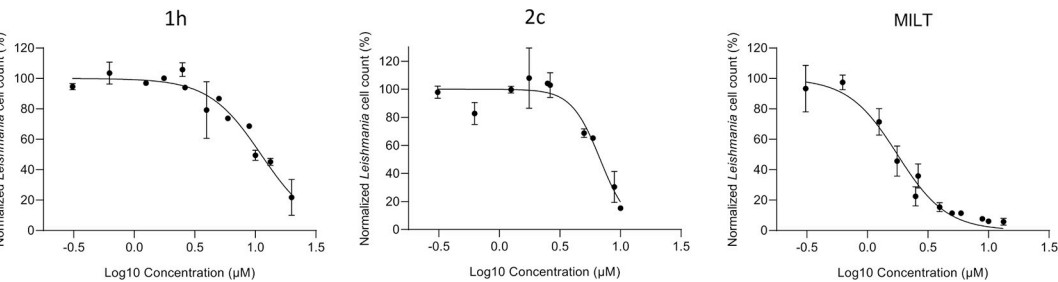

**Fig 5. Effect of 1h, 2c, and MILT on intracellular *L. infantum* amastigotes.** THP-1 cells infected with *L. infantum* MHOM/ TN/80/IPT1 and treated with **1h**, **2c** and miltefosine. In all cases cells were infected for 24 h at 37˚C; the compounds were added, and the efficacy on intracellular amastigotes was calculated after 72 h of treatment. Data are expressed as mean ± SD of two independent experiments. Each experimental condition was conducted at least in duplicate.

Consequently, four classes of more hydrophilic bisindoles, inspired by the lead URB1483, and characterized by simplification of the central pyrrole moiety, were synthesized. The compounds belonging to class **1**, the primary amines, were obtained, after basic deprotection, by microwave irradiation allowing the coupling between the indole **5** and amide **6** in a relatively short time if compared with reported methods. Alkyl- and -aryl amines and amides, classes **2** and **3**, together with the sulfonamides **4** were easily synthesized in good to excellent yields from the corresponding primary amines of class **1**.

A phenotypic screening of the 26 synthesized compounds (twenty novel) against *L. infantum* promastigotes, together with a toxicity test on human macrophage-like THP-1 cells, was performed.

All compounds, except for **1g** and **4b**, showed anti-leishmanial activity > 80% at 20 µM. In particular, classes **2** and **4**, bearing alkyl/aryl amines or amides, showed anti-promastigote activity >95%, with the exception of the *p*-nitrobenzamide derivative **3d** (90.4%). Fourteen compounds (i.e., **1d**, **1e**, **1i**, **2a**, **2b**, **2e-g**, **3a**, **3c**, **3d**, **3f**, **3g** and **4c**) showed cytotoxicity in mammalian cells (Table 2). Regarding the class **1**, the presence of heavy halides such as bromine or iodine on the indole scaffold, seems to increase the toxicity of the compounds. With the other classes **2–4**, derived from 6-bromoindole, this trend seems to continue, except for a few compounds, such as **3b** and **4a**.

Accordingly, ten active and less toxic compounds were tested on *L. infantum* promastigotes to determine $IC_{50}$. Among them, compounds **1a** and **3e** showed $IC_{50}$ >10 µM, and were not evaluated for the cytotoxicity on THP-1 cells. Compounds **1h** and **2c** appeared to be the most promising. In detail, *N*-methylated bisindole **1h** without any other substituents, showed a good $IC_{50}$ against the protozoa (4.1 µM), together with the lowest toxicity on THP 1 cells (41.6 µM), therefore resulting in the highest SI (10.1), similar to miltefosine (Table 2). On the other hand, compound **2c**, having the 6-bromobisindole scaffold with a *p*-chlorobenzyl substituent on the amine, is the most potent compound against *L. infantum* promastigotes ($IC_{50}$ = 2.7 µM), despite showing a relatively high toxicity on human cells probably due to the presence of the bromine atoms.

For these reasons, **1h** and **2c** were selected to be tested on macrophage-like THP-1 cell line infected with *L. infantum*. Compounds **1h** and **2c** also demonstrated their effectiveness in the *in vitro* infection model with $IC_{50}$ of 11.1 and 6.8 µM, respectively. The toxicity of **2c** on THP1 cells was confirmed by microscope analysis in the infection model, while **1h** showed again the better toxicity profile compared to the commercial drug miltefosine, maintaining a good activity on intracellular *L. infantum* amastigotes. Currently, the biological target of this class of molecules is still unknown, but the search is now ongoing.

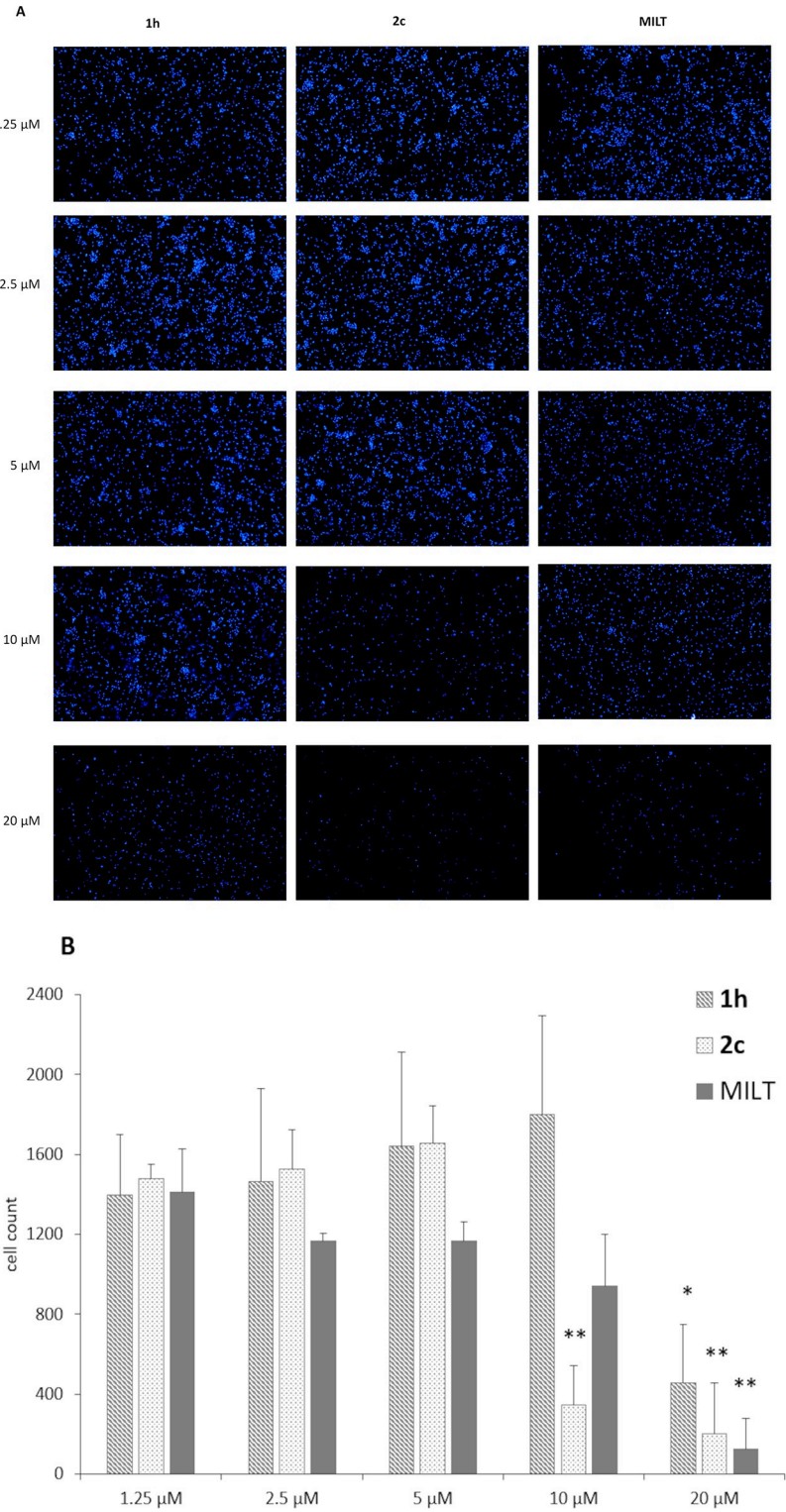

**Fig 6. Cytotoxic effect of treatment with 1h, 2c and MILT, on infected THP-1 cells after 72 h treatment.** (A) Cells were treated at concentrations of 1.25, 2.5, 5, 10, and 20 μM and stained with Hoechst dye for fluorescence microscope observation (5× magnification). (B) The effect of treatment was monitored by counting the number of cells. Data are represented as mean ± SD. Two-way ANOVA; Dunnett's multiple comparisons test; *adjusted p < 0.05, **adjusted p < 0.01.

In conclusion, **1h** shows potential to be a new hydrophilic antileishmanial agent with an acceptable cytotoxic profile. Moreover, **1h**, together with **2c**, is undoubtedly a promising lead compound for the generation of new innovative anti-infective agents.

## Supporting information

**S1 File.**
(PDF)

## Acknowledgments

Open access was funded by the Helsinki University Library. The authors thank Dr. Nina Sipari from Viikki Metabolomics Unit (Helsinki Institute of Life Science, University of Helsinki; Biocenter Finland) for her expertise with the LC-MS analyses, Prof. Francesca Ida Carducci (University of Urbino Carlo Bo) for editing the English language and style of the manuscript.

## Author Contributions

**Conceptualization:** Anabela Cordeiro-da-Silva, Paula Kiuru, Simone Lucarini, Luca Galluzzi.

**Data curation:** Aurora Diotallevi, Gloria Buffi, Diego Olivieri, Nuno Santarém, Simone Lucarini.

**Funding acquisition:** Jari Yli-Kauhaluoma, Anabela Cordeiro-da-Silva, Simone Lucarini, Luca Galluzzi.

**Investigation:** Alessia Centanni, Aurora Diotallevi, Gloria Buffi, Diego Olivieri, Nuno Santarém, Antti Lehtinen.

**Methodology:** Aurora Diotallevi, Nuno Santarém, Paula Kiuru.

**Project administration:** Simone Lucarini, Luca Galluzzi.

**Supervision:** Nuno Santarém, Anabela Cordeiro-da-Silva, Paula Kiuru, Simone Lucarini, Luca Galluzzi.

**Writing – original draft:** Aurora Diotallevi, Gloria Buffi, Diego Olivieri, Paula Kiuru, Simone Lucarini.

**Writing – review & editing:** Alessia Centanni, Aurora Diotallevi, Gloria Buffi, Diego Olivieri, Nuno Santarém, Jari Yli-Kauhaluoma, Anabela Cordeiro-da-Silva, Paula Kiuru, Simone Lucarini, Luca Galluzzi.

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
