## [Decision Letter · Decision Letter 0]

8 Feb 2024

PONE-D-24-01889Exploring hydrophilic 2,2-di(indol-3-yl)ethanamine derivatives against Leishmania infantumPLOS ONE

Dear Dr. Lucarini,

Thank you for submitting your manuscript to PLOS ONE. After careful consideration, we feel that it has merit but does not fully meet PLOS ONE’s publication criteria as it currently stands. Therefore, we invite you to submit a revised version of the manuscript that addresses the points raised during the review process.

We look forward to receiving your revised manuscript.

Kind regards,

Nisha Singh, Ph.D.

Academic Editor

PLOS ONE

Journal Requirements:

2. We note that your Data Availability Statement is currently as follows: "All relevant data are within the manuscript and its Supporting Information files."

Reviewers' comments:

Reviewer's Responses to Questions

**Comments to the Author**

1. Is the manuscript technically sound, and do the data support the conclusions?

Reviewer #1: Yes

Reviewer #2: Partly

Reviewer #3: Yes

2. Has the statistical analysis been performed appropriately and rigorously? 

Reviewer #1: Yes

Reviewer #2: Yes

Reviewer #3: Yes

3. Have the authors made all data underlying the findings in their manuscript fully available?

Reviewer #1: Yes

Reviewer #2: Yes

Reviewer #3: Yes

4. Is the manuscript presented in an intelligible fashion and written in standard English?

Reviewer #1: Yes

Reviewer #2: Yes

Reviewer #3: Yes

5. Review Comments to the Author

Reviewer #1: This is an interesting original work that describing Exploring hydrophilic 2,2-di(indol-3-yl)ethanamine derivatives against Leishmania infantum.

Following the review of the manuscript, I have no recommendations, except one: Explain how to euthanize the mice.

Reviewer #2: Dear authors, I congratulate you on your effort in creating the manuscript. However, the drug under study needs to show better protection results, and the use of a high dose of PEG is also not recommended. I am sorry, but I need to reject the manuscript.

Reviewer #3: The manuscript (PONE-D-24-01889) entitled ‘Exploring hydrophilic 2,2-di(indol-3-yl)ethanamine derivatives against Leishmania Infantum’ by prof. Simone Lucarini, Paula Kiuru and coworkers describes the design and synthesis of a new library of hydrophilic bisindole substituted analogues based on the structure of the antileishmanial compound namely URB1483, previously developed by them, which failed on the in vivo evaluation probably due to its low water solubility as the authors mentioned. In addition, the authors provide the biological evaluation of the synthesized compounds concerning against Leishmania infantum promastigotes and simultaneously for their toxicity on human macrophage-like THP-1 cells. The most active compounds 2c and 1h with IC50= 2.7 μM and IC50= 4.1 μM showed comparable activity with the oral drug miltefosine (IC50= 3.6 μM) and also comparable selectivity index only in case of 1h. The toxicity profile of the 1h was slightly better than miltefosine. On the other hand the most active compound compound 2c was more toxic than miltefosine but seems to be promising for further chemical modifications in order to achieve an improved toxicity outcome.

The manuscript overall is well written and the compounds are well characterized but some corrections are needed. These are the following:

a) I don’t fully understand the rationale related with the design of the synthesized series of compounds 2 and 3. More specifically, among the synthesized compounds included in the first series of compounds (free amine bis-indoles), 1a, 1b, 1c, 1f and 1h were found to be active against L. infantum promastigotes in the low micromolar range, with IC50s between 12.9 – 4.1μM. Among them 1a have no substitution at the aromatic ring of the NH-Indole moieties, 1b, 1c and 1f are substituted at position 6 of the NH-Indole moieties with a fluorine, chlorine or a methylester respectively and finally 1h bearing a methyl group at the indole nitrogen atoms. Since none of them bearing a 6-bromo substitution at the indole moieties and also taking into consideration that the most active was 1h with no substitution at position 6, in my opinion the authors should provide a more detailed discussion in the manuscript explaining the rationale followed for the design and synthesis of compound series 2 and 3 and mainly should explain why they proceeded only with modifications on the structure of 1d (bearing bromine atoms at 6 position of the indoles).

b) In the general procedure A used for the synthesis of bisindoles 1 the authors described that the reaction performed under microwave irradiation. Since it would be very useful for the readership of PLOS ONE, they should provide more details about the reaction condition such as the microwave power (watts) used in all the reactions performed under microwaves.

Overall, the manuscript merits publication in PLOS ONE as long as minor revision takes place according to the above suggestions.

6. PLOS authors have the option to publish the peer review history of their article (what does this mean?). If published, this will include your full peer review and any attached files.

Reviewer #1: No

Reviewer #2: No

Reviewer #3: No

---

## [Author Response · Author response to Decision Letter 0]

13 Feb 2024

Reviewer #1

Comments:

This is an interesting original work that describing Exploring hydrophilic 2,2-di(indol-3-yl)ethanamine derivatives against Leishmania infantum.

Following the review of the manuscript, I have no recommendations, except one: Explain how to euthanize the mice.

A: We thank the reviewer 1 for its supportive comments on our work. The animals were euthanised with an overdose of isoflurane followed by cervical dislocation as reported at page 4, lines 116-117 of the paper.

Reviewer #2

Comments:

Dear authors, I congratulate you on your effort in creating the manuscript. However, the drug under study needs to show better protection results, and the use of a high dose of PEG is also not recommended. I am sorry, but I need to reject the manuscript.

A: Due to the low water solubility of URB1483, 60% of PEG 400 as cosolvent was utilized to obtain a stable and injectable formulation in rats. PEG 400 is one of the most utilized cosolvent for developing formulations since it is both biocompatible and water soluble (see ref. 22-24 of the manuscript) and its concentration generally ranges from 20 to 70% in commercial injection products (Int J Pharm. 2007, 341, 1), although higher values can be also tested (Int J Pharm. 2007, 341, 1). Since in our formulation up to 60% of PEG 400 is present, the dosage is perfectly in line with literature data.

In addition, as written in the text, the aim of this research work was to find a more water-soluble analogue of URB1483, which failed the in vivo test. Indeed, we found two new hydrophilic antileishmanial agent, 1h and 2c, which could also represent promising lead compounds for the generation of new innovative and more potent anti-infective agents.

 

Reviewer #3

Comments:

The manuscript (PONE-D-24-01889) entitled ‘Exploring hydrophilic 2,2-di(indol-3-yl)ethanamine derivatives against Leishmania Infantum’ by prof. Simone Lucarini, Paula Kiuru and coworkers describes the design and synthesis of a new library of hydrophilic bisindole substituted analogues based on the structure of the antileishmanial compound namely URB1483, previously developed by them, which failed on the in vivo evaluation probably due to its low water solubility as the authors mentioned. In addition, the authors provide the biological evaluation of the synthesized compounds concerning against Leishmania infantum promastigotes and simultaneously for their toxicity on human macrophage-like THP-1 cells. The most active compounds 2c and 1h with IC50= 2.7 μM and IC50= 4.1 μM showed comparable activity with the oral drug miltefosine (IC50= 3.6 μM) and also comparable selectivity index only in case of 1h. The toxicity profile of the 1h was slightly better than miltefosine. On the other hand the most active compound compound 2c was more toxic than miltefosine but seems to be promising for further chemical modifications in order to achieve an improved toxicity outcome.

The manuscript overall is well written and the compounds are well characterized but some corrections are needed. Overall, the manuscript merits publication in PLOS ONE as long as minor revision takes place according to the following suggestions:

a) I don’t fully understand the rationale related with the design of the synthesized series of compounds 2 and 3. More specifically, among the synthesized compounds included in the first series of compounds (free amine bis-indoles), 1a, 1b, 1c, 1f and 1h were found to be active against L. infantum promastigotes in the low micromolar range, with IC50s between 12.9 – 4.1μM. Among them 1a have no substitution at the aromatic ring of the NH-Indole moieties, 1b, 1c and 1f are substituted at position 6 of the NH-Indole moieties with a fluorine, chlorine or a methylester respectively and finally 1h bearing a methyl group at the indole nitrogen atoms. Since none of them bearing a 6-bromo substitution at the indole moieties and also taking into consideration that the most active was 1h with no substitution at position 6, in my opinion the authors should provide a more detailed discussion in the manuscript explaining the rationale followed for the design and synthesis of compound series 2 and 3 and mainly should explain why they proceeded only with modifications on the structure of 1d (bearing bromine atoms at 6 position of the indoles).

A: We thank the referee for constructive comments that allow us to improve our manuscript quality. As correctly noticed, the 6-Br bisindole 1d did not show preliminary promising results as 1a-c, 1f or 1h. However, we designed the compounds library before biological test results considering the known biological activity of the privileged scaffold 1d. Indeed, classes 2 and 3 were synthetized from 1d, which is a marine bisindole alkaloid isolated from the californian tunicate Didemnum candidum and the new caledonian sponge Orina spp. Furthermore, these 6-Br bisindole derivatives could be easily functionalized by classical cross coupling reactions (e.g., Mizoroki-Heck reaction, Stille reaction, Suzuki-Miyaura reaction), therefore representing a class of useful starting materials that would easily allow the obtainment of new drug candidates.

These considerations have been added in the text at page 12, lines 380-384, together with the opportune references 30 and 31.

b) In the general procedure A used for the synthesis of bisindoles 1 the authors described that the reaction performed under microwave irradiation. Since it would be very useful for the readership of PLOS ONE, they should provide more details about the reaction condition such as the microwave power (watts) used in all the reactions performed under microwaves.

A: Thanks for the suggestion. Microwave syntheses were performed using the set temperature mode, which foresees to keep the temperature constant and the instrument automatically and consequently adjusting the power. Depending on the chosen temperature (i.e. 120°C or 80°C), the microwave power varied from 30 to 75 W. As required by the reviewer, this information has been added.

---

## [Decision Letter · Decision Letter 1]

25 Mar 2024

Exploring hydrophilic 2,2-di(indol-3-yl)ethanamine derivatives against Leishmania infantum

PONE-D-24-01889R1

Dear Dr. Lucarini,

We’re pleased to inform you that your manuscript has been judged scientifically suitable for publication and will be formally accepted for publication once it meets all outstanding technical requirements.

Kind regards,

Nisha Singh, Ph.D.

Academic Editor

PLOS ONE

Additional Editor Comments (optional):

Reviewers' comments:

Reviewer's Responses to Questions

**Comments to the Author**

1. If the authors have adequately addressed your comments raised in a previous round of review and you feel that this manuscript is now acceptable for publication, you may indicate that here to bypass the “Comments to the Author” section, enter your conflict of interest statement in the “Confidential to Editor” section, and submit your "Accept" recommendation.

Reviewer #1: All comments have been addressed

Reviewer #3: (No Response)

2. Is the manuscript technically sound, and do the data support the conclusions?

Reviewer #1: Yes

Reviewer #3: (No Response)

3. Has the statistical analysis been performed appropriately and rigorously? 

Reviewer #1: Yes

Reviewer #3: (No Response)

4. Have the authors made all data underlying the findings in their manuscript fully available?

Reviewer #1: Yes

Reviewer #3: (No Response)

5. Is the manuscript presented in an intelligible fashion and written in standard English?

Reviewer #1: Yes

Reviewer #3: (No Response)

6. Review Comments to the Author

Reviewer #1: (No Response)

Reviewer #3: (No Response)

7. PLOS authors have the option to publish the peer review history of their article (what does this mean?). If published, this will include your full peer review and any attached files.

Reviewer #1: No

Reviewer #3: No

---

## [Editor Report · Acceptance letter]

4 Jun 2024

PONE-D-24-01889R1 

PLOS ONE

Dear Dr. Lucarini, 

I'm pleased to inform you that your manuscript has been deemed suitable for publication in PLOS ONE. Congratulations! Your manuscript is now being handed over to our production team.

Kind regards, 

on behalf of

Dr. Nisha Singh 

Academic Editor

PLOS ONE